# CLASS BALANCING GAN WITH A CLASSIFIER IN THE LOOP

## ABSTRACT

Generative Adversarial Networks (GANs) have swiftly evolved to imitate increasingly complex image distributions. However, majority of the developments focus on performance of GANs on balanced datasets. We find that the existing GANs and their training regimes which work well on balanced datasets fail to be effective in case of imbalanced (i.e. long-tailed) datasets. In this work we introduce a novel and theoretically motivated Class Balancing regularizer for training GANs. Our regularizer makes use of the knowledge from a pre-trained classifier to ensure balanced learning of all the classes in the dataset. This is achieved via modelling the effective class frequency based on the exponential forgetting observed in neural networks and encouraging the GAN to focus on underrepresented classes. We demonstrate the utility of our contribution in two diverse scenarios: (i) Learning representations for long-tailed distributions, where we achieve better performance than existing approaches, and (ii) Generation of Universal Adversarial Perturbations (UAPs) in the data-free scenario for the large scale datasets, where we bridge the gap between data-driven and data-free approaches for crafting UAPs.

## 1 INTRODUCTION

Image Generation witnessed unprecedented success in recent years following the invention of Generative Adversarial Networks (GANs) by Goodfellow et al. (2014). GANs have improved significantly over time with the introduction of better architectures (Gulrajani et al., 2017; Radford et al., 2015), formulation of superior objective functions (Jolicoeur-Martineau, 2018; Arjovsky et al., 2017), and regularization techniques (Miyato et al., 2018). An important breakthrough for GANs has been the ability to effectively use the information of class conditioning for synthesizing images (Mirza & Osindero, 2014; Miyato & Koyama, 2018). Conditional GANs have been shown to scale to large datasets such as ImageNet (Deng et al., 2009) with 1000 classes (Miyato & Koyama, 2018).

One of the major issues with unconditional GANs has been their inability to produce balanced distributions over all the classes present in the dataset. This is seen as problem of missing modes in the generated distribution. A version of the missing modes problem, known as the 'covariate shift' problem was studied by Santurkar et al. (2018). One possible reason may be the absence of knowledge about the class distribution $P(Y|X)$[1] of the generated samples during training. Conditional GANs on the other hand, do not suffer from this issue since the class label $Y$ is supplied to the GAN during training. However, it has been recently found by Ravuri & Vinyals (2019) that despite being able to do well on metrics such as Inception Score (IS) (Salimans et al. (2016)) and Frèchet Inception Distance (FID) (Heusel et al., 2017), the samples generated from the state-of-the-art conditional GANs lack diversity in comparison to the underlying training datasets. Further, we observed that although conditional GANs work well in balanced case, they suffer performance degradation in the imbalanced case.

In order to address these shortcomings, we propose an orthogonal method (with respect to label conditioning) to induce the information about the class distribution $P(Y|X)$ of generated samples in the GAN framework using a pre-trained classifier. We achieve this by tracking the class distribution of samples produced by the GAN using a pre-trained classifier. The regularizer utilizes the class distribution to penalize excessive generation of samples from the majority classes, thus enforcing

---

[1]Here Y represents labels and X represents data.

the GAN to generate samples from minority classes. Our regularizer involves a novel method of modelling the forgetting of samples by GANs, based on the exponential forgetting observed in neural networks (Kirkpatrick et al. (2017)). We infer the implications of our regularizer by a theoretical bound and empirically verify the same.

We conduct empirical analysis of the proposed class balancing regularizer in two diverse and challenging scenarios:

(i) Training GANs for image generation on long-tailed datasets: Generally, even in the long-tailed distribution tasks, the test set is balanced despite the imbalance in the training set. This is because it is important to develop Machine Learning systems that generalize well across all the support regions of the data distribution, avoiding undesired over-fitting to the majority (or head) classes. Hence, it is pertinent to train GANs that can faithfully represent all classes.

(ii) Transferring the knowledge from a learnt classifier ($P(Y|X_t)$) to a GAN being trained on arbitrary prior distribution $P(X_p)$: This is a specific situation where the samples from target distribution $X_t$ are unavailable. Instead, discriminative feature knowledge is indirectly available in the form of a trained classifier ($P(Y|X_t)$). This is a perfect fit for crafting input-agnostic (Universal) adversarial perturbations in the data-free scenario. We show that the proposed regularizer can enable the generated samples to not only extract information about the target data with a trained classifier in the loop, but also represent its support to a greater extent.

In summary, our contributions can be listed as follows:

- We propose a 'class-balancing' regularizer that makes use of the statistics ($P(Y|X)$) of generated samples to promote uniformity while sampling from an unconditional GAN. The effect of our regularizer is depicted both theoretically (Section 3) and empirically (Section 4).
- We show that our regularizer enables GANs to learn uniformly across classes even when the training distribution is long-tailed. We observe gains in FID and accuracy of a classifier trained on generated samples.
- We also show that by combining a pre-trained classifier (i.e. $P(Y|X_t)$) trained on a target dataset $X_t$, with an arbitrary distribution $P(X_p)$, our framework is capable of synthesizing novel samples related to the target dataset. We show that UAPs created on such novel samples generalize to real target data and hence lead to an effective data-free attack. This application is novel to our framework and cannot be realized by conditional GANs.

## 2 BACKGROUND

### 2.1 GENERATIVE ADVERSARIAL NETWORKS (GANS)

Generative Adversarial Networks (GANs) are formulated as a two player game in which the discriminator $D$ tries to classify images into two classes: real and fake. The generator $G$ tries to generate images (transforming a noise vector $z \sim P_z$) which fool the discriminator ($D$) into classifying them as real. The game can be formulated by the following objective:

$$\min_G \max_D E_{x \sim P_r}[\log(D(x))] + E_{z \sim P_z}[\log(1 - D(G(z))] \tag{1}$$

The exact optimization for training $D$ is computationally prohibitive in large networks and the GAN is trained by alternative minimization using loss functions. Multiple loss functions have been proposed for stabilizing the GAN training. In our work we use the relativistic loss function (Jolicoeur-Martineau, 2018) which is formulated as:

$$L_D^{rel} = -E_{(x,z) \sim (P_r, P_z)}[\log(\sigma(D(x) - D(G(z))))] \tag{2}$$

$$L_G^{rel} = -E_{(x,z) \sim (P_r, P_z)}[\log(\sigma(D(G(z)) - D(x)))] \tag{3}$$

This unconditional GAN formulation does not have any class conditioning and produces different number of samples from different classes (Santurkar et al., 2018). In other words, the distribution is not balanced (uniform) across different classes for the generated data.

## 2.2 CONDITIONAL GAN

The conditional GAN (Mirza & Osindero, 2014) generates images associated with input label $y$ using the following objective:

$$\min_{G} \max_{D} E_{x \sim P_r}[\log(D(x|y))] + E_{z \sim P_z}[\log(1 - D(G(z|y)))] \tag{4}$$

The Auxillary Classifier GAN (ACGAN) (Odena et al., 2017) uses an auxiliary classifier for classification along with normal discriminator to enforce high confidence samples from the conditioned class $y$. Whereas cGAN with projection (Miyato & Koyama, 2018) uses Conditional Batch Norm (De Vries et al., 2017) in the generator and uses a projection step in the discriminator to provide class information to the GAN. We refer to this method as cGAN in the subsequent sections.

**Possible issue with Conditional GAN in Long-tailed Setting**: The objective in eq.(4) can be seen as learning a different $G(z|y)$ and $D(x|y)$ for each of the $K$ classes. In this case the tail classes with fewer samples can suffer from poor generalization as they have very few samples. However, in practice there is parameter sharing among different class generators but still class specific parameters are also present in form of Conditional BatchNorm. We find that performance of conditional GANs degrade more in comparison to unconditonal GANs in the long-tailed scenario (Section 4).

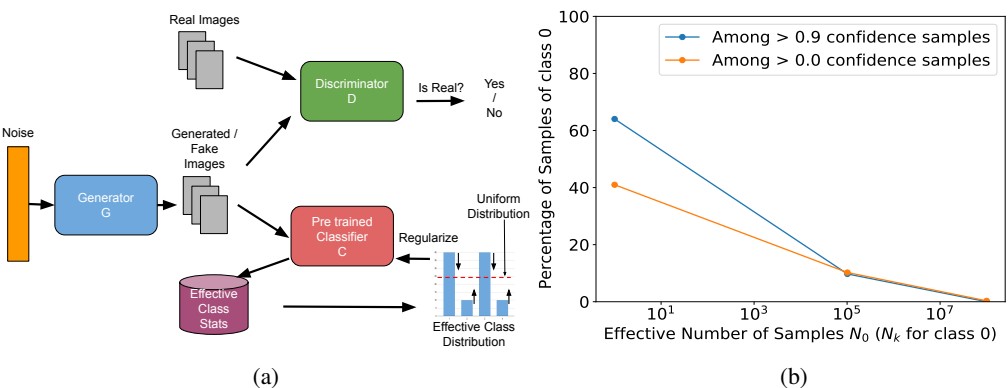

(a)                                              (b)

Figure 1: (a) shows the overview of our method and (b) shows the distribution of generated samples of SNDCGAN on CIFAR-10 for varying values of $N_0$. The percentage of class 0 (randomly choosen) samples is determined by an annotator (i.e. high accuracy classifier). When $N_0$ is large, the network tries to decrease fraction of class 0 samples whereas when $N_0$ is small it tries to increase fraction of class 0 samples among the generated samples.

## 3 METHOD

In our method we propose to introduce a pretrained classifier ($C$) to provide feedback to the generator about the label distribution $P(Y)$ over the generated images. The proposed regularizer is added with the generator loss and trained using backpropogation. We first describe the method of modelling in Section 3.1. The exact formulation of the regularizer and its theoretical properties are described in Section 3.2. The overview of our method is presented in Figure (1a).

### 3.1 CLASS STATISTICS FOR GAN

GAN is a dynamic system in which the generator $G$ has to continuously adapt itself in a way that it is able to fool the discriminator $D$. During the training, discriminator $D$ updates itself, causing the objective for the generator $G$ also to change. This change in objective can be seen as learning of different tasks for the generator $G$. In this context, we draw motivation from the seminal work on catastrophic forgetting in neural networks (Kirkpatrick et al., 2017) which shows that a neural

Figure 2: Distribution of classes and corresponding FID scores on long-tailed CIFAR-10 computed on samples generated by GANs with uniform distribution of labels in case of conditional GANs.

network trained using SGD suffers from exponential forgetting of earlier tasks when trained on a new task. Based on this, we define *effective class frequency* $\hat{N}_k^t$ of class $k$ at cycle $t$ as:

$$\hat{N}_k^t = (1 - \alpha)N_k^{\hat{t}-1} + c_k^{t-1} \tag{5}$$

Here $c_k^{t-1}$ is the number of samples of class $k$ produced by the GAN in cycle $(t-1)$. The class of the sample is determined by the pretrained classifier $C$. Although $D$ gets updated continuously, the update is slow and requires some iterations to change the form of $D$. Hence we update the statistics after certain number of iterations which compose a cycle. Here $\alpha$ is the exponential forgetting factor which is set to $0.5$ in all our experiments. We normalize the class frequency $\hat{N}_k^t$ to obtain discrete effective class distribution:

$$N_k^t = \frac{\hat{N}_k^t}{\sum_k \hat{N}_k^t} \tag{6}$$

## 3.2 REGULARIZER FORMULATION

The regularizer objective is defined as the maximization of the term($L_{reg}$) below:

$$\max_{\hat{p}} \sum_k \frac{\hat{p}_k \log(\hat{p}_k)}{N_k^t} \tag{7}$$

where $\hat{p} = \sum_{i=1}^n \frac{C(G(z_i))}{n}$. In other words, $\hat{p}$ is the average softmax vector (obtained from the classifier $C$) over the batch of $n$ samples and $\hat{p}_k$ is its $k^{th}$ component corresponding to class $k$. $z_i$ corresponds to random noise vector sampled from $P_z$. If the classifier $C$ recognizes the samples confidently with probability $\approx 1$, $\hat{p}_k$ can be seen as the approximation to the ratio of the number of samples that belong to class $k$ to the total number of samples in the batch $n$. The $N_k^t$ in the regularizer term is obtained through the update rule in Section 3.1 and is a constant during backpropagation. We want to emphasize that classifier $C$ is not required to be trained on same data as the GAN, instead it can be trained in ways such as semi-supervised learning, few-shot learning, etc. For instance, in section 4.2 we show that a classifier trained in a semi-supervised scenario also enables the GAN to produce a balanced distribution. Hence our approach doesn't specifically need labelled data which is in contrast to conditional GANs which require labels for each image while training.

**Proposition:** The maximization of the proposed objective in (7) leads to the following bound on $\hat{p}_k$:

$$\hat{p}_k \leq e^{-K(log(K)-1)\frac{N_k^t}{\sum_k N_k^t}-1} \tag{8}$$

where $K$ is the number of distinct class labels produced by classifier C. Please refer to the appendix Section A.1 for proof of the same.

**Implications of the proposition**: The bound on $\hat{p}_k$ is inversely related to the exponent of the fraction of effective class frequency $N_k^t / \sum_k N_k^t$ for a given class $k$. In case of generating a balanced distribution, $\hat{p}_k = 1/K$ which leads to the exponential average $N_k^t = 1/K$. Hence given sufficient iterations, the $\hat{p}_k$ value will achieve the upper bound which signifies tightness of the same. To demonstrate effect of the regularizer empirically, we construct two extreme case examples based on the nature of the bound:

- If $N_k^t \gg N_i^t, \forall i \neq k$, then the bound on $\hat{p}_k$ would approach $e^{-K(\log(K)-1)-1}$. Hence the network is expected to decrease the proportion of class $k$ samples.
- If $N_k^t \ll N_i^t, \forall i \neq k$, then the bound on $\hat{p}_k$ will be $e^{-1}$. Hence the network is expected to increase the proportion of class $k$ samples.

We verified the two extreme cases above by training a SNDCGAN (Miyato et al., 2018) (DCGAN with spectral normalization) on CIFAR-10 and fixing $\hat{N}_k^t$ (unnormalized version of $N_k^t$) across time steps and term it as $N_k$. Then we initialize $N_k$ to a very large value and a very small value. Results presented in Figure (1b) show that the GAN increases the proportion of samples of class $k$ in case of low $N_k$ and decreases the proportion of samples in case of large $N_k$. This shows the balancing behaviour of proposed regularizer.

## 3.3 Combining the Regularizer and GAN Objective

The regularizer is then combined with the generator loss in the following way:

$$L_g = -E_{(x,z)\sim(P_r,P_z)}[\log(\sigma(D(G(z)) - D(x)))] - \lambda L_{reg} \qquad (9)$$

It has been recently shown (Jolicoeur-Martineau, 2019) that the first term of the loss leads to minimization of $D_f(P_g, P_r)$ that is divergence between real and generated data distribution. The regularizer term ensures that the distribution of classes across generated samples is uniform. The combined objective provides insight into the working of framework, as the first term ensures that the generated images fall in the image distribution and the second term ensures that the distribution of classes is uniform. As $P_r$ comprises of diverse samples from majority class the first objective term ensures that $P_g$ is similarly diverse. The second term in the objective ensures that the discriminative properties of all classes are present uniformly in the generated distribution, which ensures that minority classes get benefit of diversity within the majority classes. This is analogous to approaches that transfer knowledge from majority to minority classes for long-tailed classifier learning (Liu et al., 2019b; Wang et al., 2017).

## 4 Experiments

For evaluating the effectiveness of our balancing regularizer, we conduct two sets of experiments: (i) image generation from long-tailed distributions, and (ii) creating Universal Adversarial Perturbations in the data-free scenario. The goal of the first task is to generate high quality images across all classes and that of the second task is to craft UAPs when the attacker has no access (e.g. due to privacy) to the target data.

### 4.1 Image Generation from long-tailed Distribution

In this experiment we aim to learn a GAN over a long-tailed dataset, which are prevalent in the real world setting. An important aspect of this problem is that it requires to transfer the knowledge from majority classes to minority classes. Several works have focused on learning classifiers for long-tailed distributions (Cao et al., 2019; Cui et al., 2019). Yet works focusing on Image Generation using long-tailed dataset are limited. Generative Minority Oversampling (GAMO) (Mullick et al., 2019) attempts to solve the problem by introducing a three player framework. We do not compare our results with GAMO as it is not trivial to extend GAMO to use schemes like Spectral Normalization, and ResGAN like architecture (Gulrajani et al., 2017) which impede fair comparison.

**Datasets**: We performed our experiments on two datasets, CIFAR-10 and a subset of LSUN. The LSUN subset consists of 250k training images and 1.5k validation images. The LSUN subset is composed of 5 balanced classes; Santurkar et al. (2018) identify this subset to be a challenging case for GANs to generate uniform distribution of classes. The original CIFAR-10 dataset is composed of 50k training images and 10k validation images. We construct the long-tailed version of the imbalanced dataset by following the same procedure as Cao et al. (2019). Here, images are removed from training dataset to convert it to a long-tailed distribution while the validation set is kept unchanged. The imbalance ratio ($\rho$) determines the ratio of number of samples in most populated class to the least populated one: $\rho = max_k\{n_k\}/min_k\{n_k\}$. More details can be found in Appendix A.2.

**Pre-Trained Classifier**: An important component of our framework is the pre-trained classifier, a ResNet32 model trained using Deffered Reweighting (DRW) of loss (Cao et al., 2019) on long-tailed versions of LSUN and CIFAR-10 datasets. Accuracy of the pre-trained classifiers and training details are present in Appendix A.3.

**GAN Architecture**: We used the SNDCGAN architecture for experiments on CIFAR-10 with images of size of $32 \times 32$ and SNResGAN (ResNet architecture with spectral normalization) structure

| Imbalance Ratio | 100 | | | | | | 10 | | | 1 |
|---|---|---|---|---|---|---|---|---|---|---|
| | FID ($\downarrow$) | KLDiv($\downarrow$) | Acc.($\uparrow$) | FID($\downarrow$) | KLDiv($\downarrow$) | Acc.($\uparrow$) | FID ($\downarrow$) |
| CIFAR-10 | | | | | | | |
| SNDCGAN | $36.97 \pm 0.20$ | $0.31 \pm 0.0$ | $68.60$ | $32.53 \pm 0.06$ | $0.14 \pm 0.0$ | $80.60$ | $27.03 \pm 0.12$ |
| ACGAN | $44.10 \pm 0.02$ | $0.33 \pm 0.0$ | $43.08$ | $38.33 \pm 0.10$ | $0.12 \pm 0.0$ | $60.01$ | $24.21 \pm 0.08$ |
| cGAN | $48.13 \pm 0.01$ | $0.02 \pm 0.0$ | $47.92$ | $26.09 \pm 0.04$ | $0.01 \pm 0.0$ | $68.34$ | $18.99 \pm 0.03$ |
| Ours | $32.93 \pm 0.11$ | $0.06 \pm 0.0$ | $72.96$ | $30.48 \pm 0.07$ | $0.01 \pm 0.0$ | $82.21$ | $25.68 \pm 0.07$ |
| LSUN | | | | | | | |
| SNResGAN | $37.70 \pm 0.10$ | $0.68 \pm 0.0$ | $75.27$ | $33.28 \pm 0.02$ | $0.29 \pm 0.0$ | $79.20$ | $28.99 \pm 0.03$ |
| ACGAN | $43.76 \pm 0.06$ | $0.39 \pm 0.0$ | $62.33$ | $31.98 \pm 0.02$ | $0.05 \pm 0.0$ | $75.47$ | $26.43 \pm 0.04$ |
| cGAN | $75.39 \pm 0.12$ | $0.01 \pm 0.0$ | $44.40$ | $30.68 \pm 0.04$ | $0.00 \pm 0.0$ | $72.93$ | $27.59 \pm 0.03$ |
| Ours | $35.04 \pm 0.19$ | $0.06 \pm 0.0$ | $77.93$ | $28.78 \pm 0.01$ | $0.01 \pm 0.0$ | $82.13$ | $28.15 \pm 0.05$ |

Table 1: Results on CIFAR-10 (top panel) and 5 class subset of LSUN (bottom panel) datasets with varying imbalance. In the last column FID values in balanced scenarios are present for ease of reference. FID, KL Div. and Acc. are calculated on 50k sampled images from each GAN.

for experiments on LSUN dataset with images size of $64 \times 64$. For the conditional GAN baselines we conditioned the generator using Conditional BatchNorm. We compare our method to two widely used conditional GANs: ACGAN and cGAN. The other baseline we use is the unconditional GAN (SNDCGAN & SNResGAN) without our regularizer. All the GANs were trained with spectral normalization in the discriminator for stabilization (Miyato et al., 2018).

**Training Setup:** We train GANs with learning rate of $0.002$ for both generator and discriminator. We used Adam optimizer with $\beta_1 = 0.5$ and $\beta_2 = 0.999$ for SNDCGAN and $\beta_1 = 0$ and $\beta_2 = 0.999$ for SNResGAN. We used a batch size of $256$ and $1$ discriminator update per generator update. As a sanity check, we use the FID values and visual inspection of images on the balanced dataset and verify the range of values from (Kurach et al., 2019). We update the statistics $N_k^t$ by update equation in Section 3.1 after every 2000 iterations. Further details are present in Appendix A.6.

**Evaluation** We used the following evaluation metrics:

**KL Divergence from Uniform Distribution of labels**: Labels for the generated samples are obtained by using the pre-trained classifier (trained on balanced data) as a proxy to annotator.
**Classification Accuracy (CA)**: We use the $\{(X, Y)\}$ pairs from the GAN generated samples to train a ResNet32 classifier and validate it on real data. For the unconditional GANs the label $Y$ is obtained from the classifier trained on long-tailed data. Note that this is similar to Classifier Accuracy Score (Ravuri & Vinyals, 2019).
**Frèchet Inception Distance (FID)**: It measures the 2-Wasserstein Distance on distributions obtained from Inception Network (Heusel et al., 2017). We use 10k samples from CIFAR-10 validation set and 10k (2k from each class) fixed random images from LSUN dataset for measuring FID.

**Discussion of Results:** We present our results in the following subsections:
1) **Stability**: In terms of stability we find that cGAN suffers from early collapse in case of high imbalance ($\rho = 100$) and stop improving under 10k iterations. Though we don't claim about instability of cGANs in general, we emphasize that the same GAN which is stable in balanced scenario is unstable in case of long-tailed version of the same dataset.
2) **Biased Distribution**: Contrary to cGAN, we find that the distribution of classes generated by ACGAN, SNDCGAN and SNResGAN becomes imbalanced. The images obtained by sampling uniformly and labelling by annotator, suffers from a high KL divergence to the uniform distribution. This leads to some classes being almost absent from the distribution of generated samples as shown in Figure 2. In this case, in Table 1 we observe FID score just differs with small margin even if there is presence of large imbalance in class distribution. Our GAN produces class samples uniformly as is evident from the low KL Divergence.
3) **Comparison with State-of-the-Art Methods**: In this work we also find that classification accuracy is weakly correlated with FID score which is in agreement to (Ravuri & Vinyals, 2019). We achieve better classifier accuracy in all cases, better than cGAN which achieves state-of-the-art Classifier Accuracy Score (CAS). Our method shows minimal degradation in FID for each long-tailed case, in comparison to the corresponding balanced case. It is also able to achieve the best FID in 3 out of 4 long-tailed cases. Hence we expect that methods such as Consistency Regularization (Zhang et al., 2019), Latent Optimization (Wu et al., 2019b) etc. can be applied in conjunction with our method to further improve the quality of images. But in this work we specifically focused on techniques used to provide class information $Y$ of an image $X$ to the GAN. Several state-of-the-art GANs use an approach similar to cGAN (Wu et al., 2019b; Brock et al., 2018) for conditioning the discriminator and the generator.

| | FID ($\downarrow$) | KLDiv($\downarrow$) | FID ($\downarrow$) | KLDiv($\downarrow$) |
|---|---|---|---|---|
| Imbalance Ratio | 100 | | 10 | |
| CIFAR-10 | | | | |
| SNDCGAN | $36.97 \pm 0.20$ | $0.31 \pm 0.0$ | $32.53 \pm 0.06$ | $0.14 \pm 0.0$ |
| Ours (Supervised) | $32.93 \pm 0.11$ | $0.06 \pm 0.0$ | $30.48 \pm 0.07$ | $0.01 \pm 0.0$ |
| Ours (Semi Supervised) | $33.32 \pm 0.03$ | $0.14 \pm 0.0$ | $30.37 \pm 0.14$ | $0.04 \pm 0.0$ |
| LSUN | | | | |
| SNResGAN | $37.70 \pm 0.10$ | $0.68 \pm 0.0$ | $33.28 \pm 0.02$ | $0.29 \pm 0.0$ |
| Ours (Supervised) | $35.04 \pm 0.19$ | $0.06 \pm 0.0$ | $28.78 \pm 0.01$ | $0.01 \pm 0.0$ |
| Ours (Semi Supervised) | $35.95 \pm 0.05$ | $0.15 \pm 0.0$ | $30.96 \pm 0.07$ | $0.06 \pm 0.0$ |

Table 2: Comparison of results in Semi Supervised Setting. The pretrained classifier used in our framework is fine-tuned with 0.1% of labelled data. The same classifier trained on balanced dataset is used as annotator for calculating KL Divergence for all baselines.

| | FID ($\downarrow$) | KLDiv($\downarrow$) |
|---|---|---|
| SNResGAN | $30.05 \pm 0.05$ | $0.18 \pm 0.0$ |
| ACGAN | $69.90 \pm 0.13$ | $0.40 \pm 0.0$ |
| cGAN | $30.87 \pm 0.06$ | $0.09 \pm 0.0$ |
| Ours | $28.17 \pm 0.06$ | $0.11 \pm 0.0$ |

Table 3: Results on long-tailed CIFAR-100 dataset with imbalance ratio = 10. FID is computed through 50k generated images and KL Div of class distribution of GAN and uniform distribution is present in last column.

We also find that our method trained using SNResGAN performs similarly to experiments in Table 1 on long-tailed CIFAR-100 dataset as well. Our method achieves the best FID of 28.17 among all baselines and also achieves balanced class distribution like cGAN. The results are summarized in Table 3 and detailed experimental details are present in Appendix A.6.1.

## 4.2 Semi-supervised class-balancing GAN

In this section we show that the presence of classifier in our framework is an advantage for it, as it allows classifiers trained through various sources to be used for providing feedback to GAN. This feedback allows the GAN to generate class balanced distributions in cases when the labels for underlying long-tailed distributions are not known. This reduces the need of labelled data in our framework and shows the effectiveness over conditional GAN. As it has been shown in that performance of conditional GANs deteriote (Lucic et al., 2019) when used with limited labelled data. We use a ResNet-50 pretrained model on ImageNet from BiT (Big Image Transfer) (Kolesnikov et al., 2019) and fine tune it using 0.1 % of labelled data of balanced training set (i.e. 5 images per class for CIFAR-10 and 50 images per class for LSUN dataset). In all long-tailed cases this amount of data for each class is present in the training set.

We observe that with just using 0.1% labelled data we are able to obtain a significantly balanced distribution as seen by low KL Divergence in comparison to unconditonal GAN (in Table 2) and also achive better FID score than unsupervised GAN. This application is unique to our framework as conditional GANs explicity require labels for whole dataset for training. The experimental details are present in Appendix A.6.

## 4.3 Data-Free Universal Adversarial Perturbation

Adversarial perturbation (Szegedy et al. (2013)) is a structured noise added to a benign data sample with an aim of confusing the machine learning model processing it, leading to an inaccurate inference. Universal Adversarial Perturbations (UAP) (Moosavi-Dezfooli et al. (2017)) are such noises that are input agnostic and can fool the model when added to any data sample. These perturbations demonstrate transferability across different deep CNN models posing a challenge to their deployability. Crafting UAPs require original training data on which the target deep model is trained. However, the dataset access can be limited due to privacy restrictions. Attackers overcome this limitation via (i) formulating data-free objectives (e.g. Mopuri et al. (2017)), or (ii) using a proxy dataset composed of either arbitrary natural samples (e.g. Zhang et al. (2020)) or generated synthetic samples (e.g. Mopuri et al. (2018b)). GAN inspired generative modelling (Poursaeed et al. (2018); Mopuri et al. (2018a;b)) of the UAPs for a given CNN classifier has been shown to capture these input agnostic vulnerabilities. However, in the absence of the target training data, these models suffer from lack of knowledge about the training distribution. Further, synthetic samples generated using existing methods (e.g. Mopuri et al. (2018b)) lack diversity and use an activation maximization approach which is computationally expensive since optimization has to be performed for each batch of samples separately. To tackle this issue, we introduce an activation maximization term in our GAN objective to combine discriminative class knowledge ($P(Y|X_t)$) learnt by the classifier

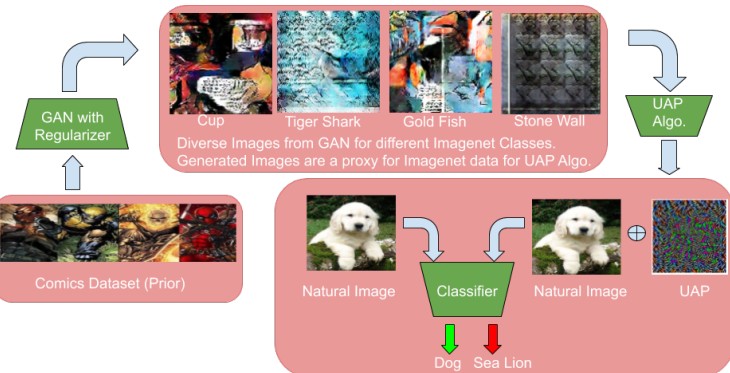

Figure 3: Overview of our UAP crafting approach using arbitrary data and a classifier in the loop. First, with the help of the proposed regularizer, the GAN can generate samples enriched with features from the ImageNet pre-trained classifier. The UAP algorithm can subsequently craft better perturbations due to the available discriminative information about the target (ImageNet) data.

$(C)$ trained on target data $(X_t)$ with an arbitrary prior distribution $P(X_p)$. We present an overview of our approach in Figure 3.

In the absence of the target data on which the victim CNN classifier is trained, we first train a GAN on an arbitrary dataset. Through our regularizer, we encourage the GAN to generate samples from all the modes of the target data. This is achieved by incorporating the pre-trained CNN classifier in the optimization as discussed in Section 3. Once the GAN is trained, we use the generated samples as a proxy to the target data for crafting UAPs. Since these samples represent the support of the target data modes, they bring in useful prior about the same, enabling the attacker to craft effective UAPs. In the UAP experiments we use Comics Dataset (Comics-Dataset) as the arbitrary prior $P(X_p)$ and use the ResNet-18 (He et al., 2016) classifier trained on ImageNet Deng et al. (2009) to impart class specific features through the activation maximization loss. However the use of Activation maximization (AM) alone with GAN can not encourage GAN to learn features of multiple target classes (i.e. modes). This issue is resolved by making use of our regularizer which encourages the GAN to learn different modes. The final generator objective can then be written as:

$$L_g = L_G^{rel} - \lambda L_{reg} + L_{AM} \tag{10}$$

$$L_{AM} = E_{z \sim P_z}[H(C(G(z)))] \tag{11}$$

where $H(C(G(z)))$ is the entropy of the classifier output for the generated data. This application is unique to our framework and cannot be realized by other conditional GANs. We use a DCGAN architecture to generate $128 \times 128$ images using a prior distribution of comic images (Comics-Dataset). It is found that (Odena et al. (2017)) generating a large number of classes is difficult for a single DCGAN even with conditioning. However, with the proposed regularizer, we are able to generate samples which are classified into a very diverse set of 968 ImageNet classes by ResNet-18 classifier, whereas just using Activation Maximization with GAN resulted in limited set of 25 labels. We also find that diversity in classes helps a lot in improving the fooling rate for which ablation results are present in Table 5. The regularizer in each cycle encourages GAN to shift it's focus to the underrepresented classes. Due to the limited capacity of DCGAN it's bound to forget some classes due to shift in focus caused by regularizer. For mitigating this we sample images from multiple cycles the exact details of procedure are described in Appendix A.7. The above procedure was adequate for our experiments, for resolving the forgetting issues in DCGAN large capacity architectures from BigGAN (Brock et al., 2018) can be used. The exact hyperparameters and architecture details are present in the Appendix A.7.

**UAP Generation and Results:** We use Generative Adversarial Perturbation (Poursaeed et al., 2018) which is an off the shelf algorithm for training a generator $G$ for crafting UAPs. We also allow the gradients to flow to deeper ResNet layers using the method introduced by (Wu et al., 2019a). We replace the ImageNet training data with the prior images generated by the GAN described above. We find that a single GAN with ResNet-18 network is enough to generate effective priors for fooling several ImageNet models. For evaluation we follow existing works and limit the strength of perturbation to $\ell_\infty = 10$. We report Fooling Rate (FR), which is the percentage of data samples for which addition of our UAP flips the predicted label. We use $-\log(H(C(x), y_x))$ (i.e negative of log cross

| Method | VGG-16 | VGG-19 | ResNet-50 | ResNet-152 | Mean FR |
|---|---|---|---|---|---|
| GDUAP + P (Mopuri et al., 2019) | 64.95 | 52.49 | 56.70 | 44.23 | 53.89 |
| PD-UA + P (Liu et al., 2019a) | 70.69 | 64.98 | 63.50 | 46.39 | 60.69 |
| AAA (Mopuri et al., 2018b) | 71.59 | 72.84 | - | 60.72 | 68.38 |
| MI-ADV* (Zhang et al., 2020) | 92.20 | 91.60 | - | 79.90 | 87.9 |
| Ours | 96.16 | 94.73 | 83.72 | 94.00 | **94.96** |
| MI-ADV** (With ImageNet) | 94.30 | 94.98 | - | 90.08 | 93.12 |

Table 4: Comparison of our UAP performance (Fooling Rate) to the state-of-the-art approaches. The Mean FR is the mean of VGG-16, VGG-19 and ResNet-152 as those are provided by all other approaches. * These results use MSCOCO (Lin et al., 2014) as prior distribution which overlaps with the target ImageNet categories. **These results are with using the target ImageNet data itself (i.e. in the presence of the data on which the victim classifier is trained).

| Prior | Fooling Rate |
|---|---|
| Comics | 49.66 |
| GAN + AM | 63.89 |
| Ours | 83.72 |
| ImageNet Data* | **89.11** |

Table 5: Ablation on Different Priors for ResNet-50 model. *For ImageNet we find that $-H(C(x), y_x)$ (i.e. negative cross entropy) is more effective hence we report the better fooling rate.

entropy) as the fooling loss as prescribed by (Poursaeed et al., 2018) for all networks. The detailed results are presented in Table 4. Note that our data free results are better than not only the existing data-free approaches by a large margin but also the recent data-driven method (Zhang et al., 2020) which uses ImageNet training data by a considerable 2%. We provide a detailed comparison of the data used by various approaches in Appendix A.7.

## 5 DISCUSSION

In this section we discuss some of the important aspects of our work:

- Our approach can be directly applied for semi supervised GAN learning as it decouples classifier learning and data which can enable learning on unlabeled data.
- We would like to emphasize that the presence of a classifier in our framework is not a disadvantage. There has been significant progress in classification setup in semi supervised learning and learning from long-tailed distributions. We also show that classifiers obtained from such methods can also be used in our framework in Section 4.2.
- We have noticed while training GAN for the UAP application, on multiple occasions, that texture alone is transferred as a discriminative feature from the classifier. This may be due to the bias of the classifiers towards texture (Geirhos et al., 2018) and image generation will improve as the classifiers improve. However, it still serves as an effective prior about the modes (classes) in the underlying data distribution on which the classifiers are trained.
- The class balancing problem differs from data coverage problem (Yu et al., 2020; Srivastava et al., 2017) as the latter tends to make the generated distribution similar to data distribution. Training on long-tailed data can induce the GAN distribution to be long-tailed as well.

## 6 CONCLUSION

In this paper, we propose a class-balancing regularizer to balance class distribution of generated samples while training GANs. We present its implications in terms of a theoretical bound and comprehensive experimental analysis in case of long-tailed data distributions. We have demonstrated the utility of our regularizer beyond the GAN framework in crafting input agnostic adversarial perturbations. The effectiveness of our contribution is exhibited through state-of-the-art performance on training of GANs on long-tailed data distributions as well as in crafting Universal Adversarial Perturbations in a data-free setting.

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

## A APPENDIX

### A.1 PROOF OF THE PROPOSITION

**Proposition:** The proposed objective below:

$$\max_{\hat{p_k}} \sum_k \frac{\hat{p_k} \log(\hat{p_k})}{N_k^t} \tag{12}$$

leads to thefollowing bound on $\hat{p_k}$:

$$\hat{p_k} \leq e^{-K(\log(K)-1)\frac{N_k^t}{\sum_k N_k^t}-1} \tag{13}$$

where $K$ is the number of distinct class labels produced by classifier C.

**Proof:**

$$\max_{\hat{p_k}} \sum_k \frac{\hat{p_k} \log(\hat{p_k})}{N_k^t} \tag{14}$$

Introducing the probability constraint and the Lagrange multipltiplier $\lambda$:

$$L(\hat{p}, \lambda) = \sum_k \frac{\hat{p_k} \log(\hat{p_k})}{N_k^t} - \lambda(\sum \hat{p_k} - 1) \tag{15}$$

On solving the equations obtained by setting $\frac{\partial L}{\partial \hat{p_k}} = 0$ :

$$\frac{1}{N_k^t} + \frac{\log(\hat{p_k})}{N_k^t} - \lambda = 0 \implies \hat{p_k} = e^{\lambda N_k^t - 1} \tag{16}$$

Using the constraint $\dfrac{\partial L}{\partial \lambda} = 0$ we get:

$$\sum_k \hat{p_k} = 0 \implies \sum_k e^{\lambda N_k^t - 1} = 1 \implies \sum_k e^{\lambda N_k^t} = e \tag{17}$$

Now we normalize both sides by $1/K$ where K is the distinct labels produced by classifier and apply Jensen's inequality for concave function $\psi(\frac{\sum a_i x_i}{\sum a_i}) \geq \frac{\sum a_i \psi(x_i)}{\sum a_i}$ and use $\psi$ as log function:

$$\frac{e}{K} = \sum_k \frac{e^{\lambda N_k^t}}{K} \implies \log(\frac{e}{K}) = \log(\sum_k \frac{e^{\lambda N_k^t}}{K}) \geq \sum_k \frac{\lambda N_k^t}{K} \tag{18}$$

On substituting the value of $\lambda$ in inequality:

$$K(1 - \log(K)) \geq \lambda \sum_k N_k^t \implies K(1 - \log(K)) \geq (\sum_k N_k^t)\frac{1 + \log(\hat{p_k})}{N_k^t} \tag{19}$$

On simplifying and exponentiation we get the following result:

$$\hat{p_k} \leq e^{-K(\log(K)-1)\frac{N_k^t}{\sum_k N_k^t} - 1} \tag{20}$$

We observe that the penalizing factor $K(\log(K) - 1)$ is increasing in terms of number of classes $K$ in the dataset which is advantageous to us as we need a large penalizing factor as $N_k^t / \sum_k N_k^t$ will be smaller when number of classes is large in the dataset.

## A.2 DATASETS

We use CIFAR-10 (Krizhevsky et al., 2009) dataset for our experiments which has 50k training images and 10k validation images. For the LSUN (Yu et al., 2015) dataset we use a fixed subset of 50k training images for each of bedroom, conference room, dining room, kitchen and living room classes. In total we have 250k training images and 1.5k validation set of images for LSUN dataset. The imbalanced versions of the datasets are created by removing images from the training set.

## A.3 PRE TRAINED CLASSIFIER DETAILS

All the pre-trained classifiers used for Image generation experiments use a ResNet32(He et al., 2016) classifier. The classifier is trained using Deferred Re-weighting (DRW) scheme Cao et al. (2019); Cui et al. (2019) with effective number of samples. We use the open source code available at https://github.com/kaidic/LDAM-DRW. We use the same learning rate schedule of initial learning rate of 0.01 and multiplying by 0.01 at epoch 160 and 180. We train the models for 200 epochs and start reweighting at epoch 160. We give a summary of the validation accuracy of the models in the following table: The classifier obtained by training on the balanced scenario is used as an annotator

| Imbalance Ratio | 100 | 10 | 1 |
|---|---|---|---|
| CIFAR-10 | 76.67 | 87.70 | 92.29 |
| LSUN | 82.40 | 88.07 | 90.53 |

Table 6: Validation Accuracy of the PreTrained Classifiers used with GAN's. The balanced classifier also serves as an annotator.

for obtaining class labels for GAN generated samples. We use the same ResNet32 (He et al., 2016) classifier with the same learning rate schedule as above with cross entropy loss to obtain Classifier Accuracy.

## A.4 ARCHITECTURE DETAILS FOR GAN

We use the SNDCGAN architecture for experiments on CIFAR-10 and SNResGAN architecture for experiments on LSUN dataset Gulrajani et al. (2017); Miyato et al. (2018). The notation for the architecture tables are as follows: m is the batch size, FC(dim_in, dim_out) is a fully connected Layer,

CONV(channels_in, channels_out, kernel_size, stride) is convolution layer, TCONV(chanels_in, channel_out, kernel_size, stride) is the transpose convolution layer, BN is BatchNorm (Ioffe & Szegedy, 2015) Layer in case of unconditonal GANs and conditional BatchNorm in case of conditional GANs. LRelu is the leaky relu activation function and GSP is the Global Sum Pooling Layer. The DIS_BLOCK(channels_in, channels_out, downsampling) and GEN_BLOCK(channels_in, channels_out, upsampling) correspond to the Discriminator and Generator block used in the (Gulrajani et al., 2017). The architectures are presented in detail in Tables 7, 8, 9 and 10.

| Layer | Input | Output | Operation |
|---|---|---|---|
| Input Layer | (m, 128) | (m, 8192) | FC(128, 8192) |
| Reshape Layer | (m, 8192) | (m, 4, 4, 512) | RESHAPE |
| Hidden Layer | (m, 4, 4, 512) | (m, 8, 8, 256) | TCONV(512, 256, 4, 2),BN,LRELU |
| Hidden Layer | (m, 8, 8, 256) | (m, 16, 16, 128) | TCONV(256, 128, 4, 2),BN,LRELU |
| Hidden Layer | (m, 16, 16, 128) | (m, 32, 32, 64) | TCONV(128, 64, 4, 2),BN,LRELU |
| Hidden Layer | (m, 32, 32, 64) | (m, 32, 32, 3) | CONV(64, 3, 3, 1) |
| Output Layer | (m, 32, 32, 3) | (m, 32, 32, 3) | TANH |

Table 7: Generator of SNDCGAN (Miyato et al., 2018; Radford et al., 2015) used for CIFAR10 image synthesis.

| Layer | Input | Output | Operation |
|---|---|---|---|
| Input Layer | (m, 32, 32, 3) | (m, 32, 32, 64) | CONV(3, 64, 3, 1), LRELU |
| Hidden Layer | (m, 32, 32, 64) | (m, 16, 16, 64) | CONV(64, 64, 4, 2), LRELU |
| Hidden Layer | (m, 16, 16, 64) | (m, 16, 16, 128) | CONV(64, 128, 3, 1), LRELU |
| Hidden Layer | (m, 16, 16, 128) | (m, 8, 8, 128) | CONV(128, 128, 4, 2), LRELU |
| Hidden Layer | (m, 8, 8, 128) | (m, 8, 8, 256) | CONV(128, 256, 3, 1), LRELU |
| Hidden Layer | (m, 8, 8, 256) | (m, 4, 4, 256) | CONV(256, 256, 4, 2), LRELU |
| Hidden Layer | (m, 4, 4, 256) | (m, 4, 4, 512) | CONV(256, 512, 3, 1), LRELU |
| Hidden Layer | (m, 4, 4, 512) | (m, 512) | GSP |
| Output Layer | (m, 512) | (m, 1) | FC(512, 1) |

Table 8: Discriminator of SNDCGAN (Miyato et al., 2018) used for CIFAR10 image synthesis.

| Layer | Input | Output | Operation |
|---|---|---|---|
| Input Layer | (m, 128) | (m, 16384) | FC(128, 16384) |
| Reshape Layer | (m, 16384) | (m, 4, 4, 1024) | RESHAPE |
| Hidden Layer | (m, 4, 4, 1024) | (m, 8, 8, 512) | GEN_BLOCK(1024, 512, True) |
| Hidden Layer | (m, 8, 8, 512) | (m, 16, 16, 256) | GEN_BLOCK(512, 256, True) |
| Hidden Layer | (m, 16, 16, 256) | (m, 32, 32, 128) | GEN_BLOCK(256, 128, True) |
| Hidden Layer | (m, 32, 32, 128) | (m, 64, 64, 64) | GEN_BLOCK(128, 64, True) |
| Hidden Layer | (m, 64, 64, 64) | (m, 64, 64, 3) | BN, RELU, CONV(64, 3, 3, 1) |
| Output Layer | (m, 64, 64, 3) | (m, 64, 64, 3) | TANH |

Table 9: Generator of SNResGAN used for LSUN image synthesis.

| Layer | Input | Output | Operation |
|-------|-------|--------|-----------|
| Input Layer | (m, 64, 64, 3) | (m, 32, 32, 64) | DIS_BLOCK(3, 64, True) |
| Hidden Layer | (m, 32, 32, 64) | (m, 16, 16, 128) | DIS_BLOCK(64, 128, True) |
| Hidden Layer | (m, 16, 16, 128) | (m, 8, 8, 256) | DIS_BLOCK(128, 256, True) |
| Hidden Layer | (m, 8, 8, 256) | (m, 4, 4, 512) | DIS_BLOCK(256, 512, True) |
| Hidden Layer | (m, 4, 4, 512) | (m, 4, 4, 1024) | DIS_BLOCK(512, 1024, False), RELU |
| Hidden Layer | (m, 4, 4, 1024) | (m, 1024) | GSP |
| Output Layer | (m, 1024) | (m, 1) | FC(1024, 1) |

Table 10: Discriminator of SNResGAN (Miyato et al., 2018; Gulrajani et al., 2017) used LSUN for image synthesis.

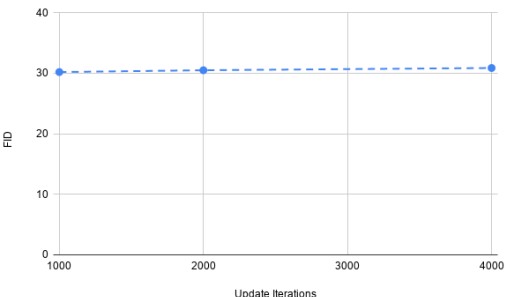

Figure 4: Effect on FID with change in num steps for statistics update(For CIFAR-10 imbalance ratio $\rho = 10$)

## A.5   HYPERPARAMETER CONFIGURATION (IMAGE GENERATION EXPERIMENTS)

### A.5.1   LAMBDA THE REGULARIZER COEFFECIENT

The $\lambda$ hyperparameter is the only hyperparameter that we change across different imbalance scenarios. As the overall objective is composed of the two terms:

$$L_g = -E_{(x,z)\sim(P_r,P_z)}[\log(\sigma(D(G(z)) - D(x))] - \lambda L_{reg} \tag{21}$$

As the number of terms in the regularizer objective can increase with number of classes $K$. For making the regularizer term invariant of $K$ and also keeping the scale of regularizer term similar to GAN loss, we normalize it by $K$. Then the loss is multiplied by $\lambda$. Hence the effective factor that gets multiplied with regularizer term is $\frac{\lambda}{K}$.

| Imbalance Ratio ($\rho$) | 100 | 10 | 1 |
|---|---|---|---|
| CIFAR-10 | 10 | 7.5 | 5 |
| LSUN | 20 | 7.5 | 5 |

Table 11: Values of $\lambda$ for different imbalance cases. For LSUN the $\lambda$ gets divide by 5 and for $\lambda$ it gets divided by 10 before multiplication to regularizer term.

The presence of pre-trained classifier which provides labels for generated images makes it easy to determine the value of $\lambda$. Although the pre-trained classifier is trained on long-tailed data its label distribution is sufficient to provide a signal for balance in generated distribution. We use the KL Divergence of labels with respect to uniform distribution for 10k samples in validation stage to check for balance in distribution and choose $\lambda$ accordingly. We use the FID implementation available here [2].

### A.5.2   OTHER HYPERPARMETERS

We update the effective class distribution periodically after 2k updates (i.e. each cycle defined in section 3 consists of 2k iteration). We find the algorithm performance to be stable for a large range of update frequency depicted in Figure 4. We also apply Exponential Moving Average on generator weights after 20k steps for better generalization. The hyperparameters are present in detail in Table 12. **Validation Step:** We obtain the FID on 10k generated samples after each 2k iterations and choose the checkpoint with best FID for final sampling and FID calulation present in Table 1. **Convergence of Network**: We find that our GAN + Regularizer setup also achives similar convergence in FID value to the GAN without the regularizer. We show the FID curves for the CIFAR-10 (Imbalance Ratio = 10) experiments in Figure 5.

## A.6   HYPERPARAMETERS FOR THE SEMI SUPERVISED GAN ARCHITECTURE

We use a ImageNet and ImageNet-21k pre-trained model with ResNet 50 architecutre as the base model. The fine tuning of the model on CIFAR-10 and LSUN has been done by using the code of

---

[2]https://github.com/mseitzer/pytorch-fid

| Parameter | Values(CIFAR-10) | Values(LSUN) |
|---|---|---|
| Iterations | 100k | 100k |
| Generator lr | 0.002 | 0.002 |
| Discriminator lr | 0.002 | 0.002 |
| Adam ($\beta_1$) | 0.5 | 0.0 |
| Adam ($\beta_2$) | 0.999 | 0.999 |
| Batch Size | 256 | 256 |
| EMA(Start After) | 20k | 20k |
| EMA(Decay Rate) | 0.9999 | 0.9999 |

Table 12: Hyperparameter Setting for Image Generation Experiments.

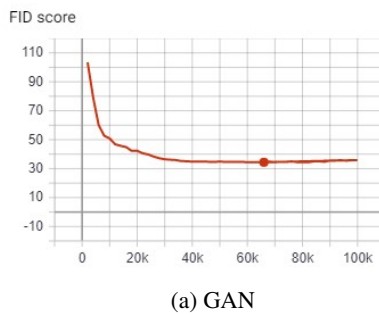

(a) GAN

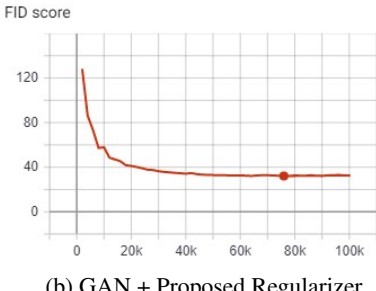

(b) GAN + Proposed Regularizer

Figure 5: Plots of FID (y axis) vs Number of iteration steps. We observe a similar curve in both the cases.

notebook present here [3]. The accuracy of the classifiers fine-tuned on validation data, trained with 0.1% of labelled data is 84.96 % for CIFAR-10 and 82.40 % for LSUN respectively. The lambda (regularizer coeffecient) values are present in the table below:

| Imbalance Ratio ($\rho$) | 100 | 10 |
|---|---|---|
| CIFAR-10 | 10 | 7.5 |
| LSUN | 10 | 7.5 |

Table 13: Values of $\lambda$ for different imbalance cases. For LSUN the $\lambda$ gets divide by 5 and for $\lambda$ it gets divided by 10 before multiplication to regularizer term.

The training hyper parameters are same as the ones present in the Table 12. Only in case of LSUN semi supervised experiments we use a batch size of 128 to fit into GPU memory for semi supervised experiments.

### A.6.1 RESULTS ON CIFAR-100

In this section we show results on CIFAR-100 dataset which has 100 classes having 500 images for each class. We use SNResGAN architecture from Miyato & Koyama (2018), which is similar to SNResGAN architecure used for LSUN experiments. The architecutre is used for generating $32 \times 32$ images. We use the same hyperparameters used for LSUN experiments listed in Table 12. We use a $\lambda$ value of 0.5 for CIFAR-100 experimets. The results in Table 3 show that our method on long-tailed CIFAR100 of using GAN + Regularizer achieves the best FID and also have class balance similar to cGAN (conditional GAN). The labels for the samples generated by GAN are obtained by a classfier trained on balanced CIFAR-100 dataset. The KL Divergence between the GAN label distribution and uniform distribution is present in Table 3. The classifier for obtaining class labels for KL Divergence evaluation is trained on balanced CIFAR-100 with setup described in A.3 which serves as annotator for all methods.

---

[3]https://github.com/google-research/big_transfer/blob/master/colabs/big_transfer_pytorch.ipynb

## A.7 UAP EXPERIMENTAL DETAILS

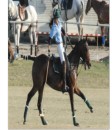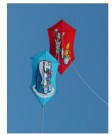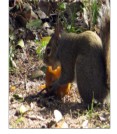 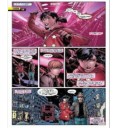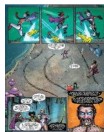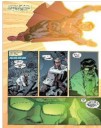

(a) Samples from COCO dataset  (b) Samples from Comics Dataset

Figure 6: Comparison of samples used by our approach vs approach by Zhang et al. (2020)

**Dataset**: We use the Comics dataset (Comics-Dataset) whereas approach (Zhang et al., 2020) use COCO dataset. COCO dataset has overlap with ImageNet categories. The difference in images used shows that our procedure does not require natural images for generating effective attack. This increases applicability of our method. We use a DCGAN architecture to generate $128 \times 128$ images from the GAN described in Table 15 and 16. In this experiment, we update the mode statistics after every epoch. Hyperparameters are present in the table 14. The images generated by our method is present in figure 10.

| Parameter | Value |
|---|---|
| Iterations | 200 epochs (33.6k) |
| Generator lr | 0.002 |
| Discriminator lr | 0.002 |
| Adam ($\beta_1$) | 0.5 |
| Adam ($\beta_2$) | 0.999 |
| $\lambda$ (Regularizer) | 2 |
| Batch Size | 512 |

Table 14: Hyperparameters for DCGAN.

**Sampling:** We find that sampling in different cycles produces samples from diverse classes as due to our regularizer as it enforces learning of different underrepresented classes in different cycles. Hence we sample 1024 images each from the GAN checkpoints in the last 40 cycles to obtain the dataset for UAP generation. This also shows that the regularizer is effective in shifting the distribution of GAN to produce different modes which is not possible with just Activation Maximization (AM). The increase in number of diverse classes is shown in Figure 7.

**Generative Adversarial Perturbations:** We use the author's Pytorch implementation of the algorithm to generate attacks. For ResNets we allow the gradients to pass through skip connections by using the method of Wu et al. (2019a) with $\alpha = 0.5$. We train the algorithm for 20 epochs in each case except in case of VGG16. For VGG16 we use an additional factor of 10 with the loss to make fooling rate converge in 20 epochs.

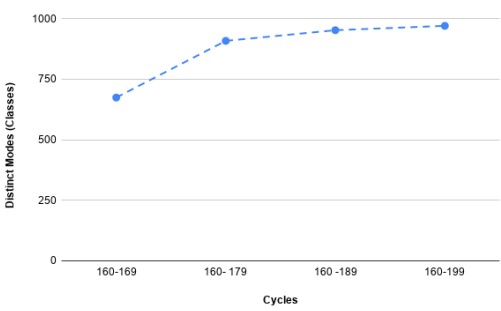

Figure 7: Number of distinct Modes observed while sampling from different cycles

| Layer | Input | Output | Operation |
|---|---|---|---|
| Input Layer | (m, 100) | (m, 4, 4, 1024) | TConv(100, 1024, 4, 1) |
| Hidden Layer | (m, 4, 4, 1024) | (m, 8, 8, 512) | TConv(1024, 512, 4, 2),BN,LRelu |
| Hidden Layer | (m, 8, 8, 512) | (m, 16, 16, 256) | TConv(256, 128, 4, 2),BN,LRelu |
| Hidden Layer | (m, 16, 16, 256) | (m, 32, 32, 128) | TConv(256, 128, 4, 2),BN,LRelu |
| Hidden Layer | (m, 32, 32, 128) | (m, 64, 64, 64) | TConv(128, 64, 4, 2),BN,LRelu |
| Hidden Layer | (m, 64, 64, 64) | (m, 128, 128, 3) | TConv(64, 3, 4, 2),BN,LRelu |
| Output Layer | (m, 128, 128, 3) | (m, 128, 128, 3) | Tanh |

Table 15: Generator of DCGAN (Radford et al., 2015) used for UAP Experiments.

| Layer | Input | Output | Operation |
|---|---|---|---|
| Input Layer | (m, 128, 128, 3) | (m, 64, 64, 64) | Conv(3, 64, 4, 2), LRelu |
| Hidden Layer | (m, 64, 64, 64) | (m, 32, 32, 128) | Conv(64, 128, 4, 2),BN, LRelu |
| Hidden Layer | (m, 32, 32, 128) | (m, 16, 16, 256) | Conv(128, 256, 4, 2),BN, LRelu |
| Hidden Layer | (m, 16, 16, 256) | (m, 8, 8, 512) | Conv(256, 512, 4, 2),BN, LRelu |
| Hidden Layer | (m, 8, 8, 512) | (m, 4, 4, 1024) | Conv(512, 1024, 4, 2),BN, LRelu |
| Output Layer | (m, 4, 4, 1024) | (m, 1) | Conv(1024, 1, 4, 1), LRelu |

Table 16: Discriminator of DCGAN (Radford et al., 2015) used for UAP Experiments.

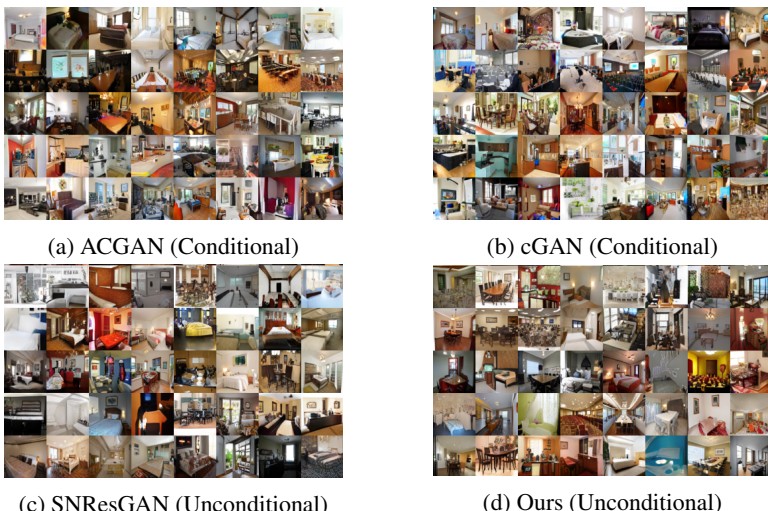

(a) ACGAN (Conditional)
(b) cGAN (Conditional)
(c) SNResGAN (Unconditional)
(d) Ours (Unconditional)

Figure 8: Images from different GANs with imbalance ratio ($\rho = 10$)

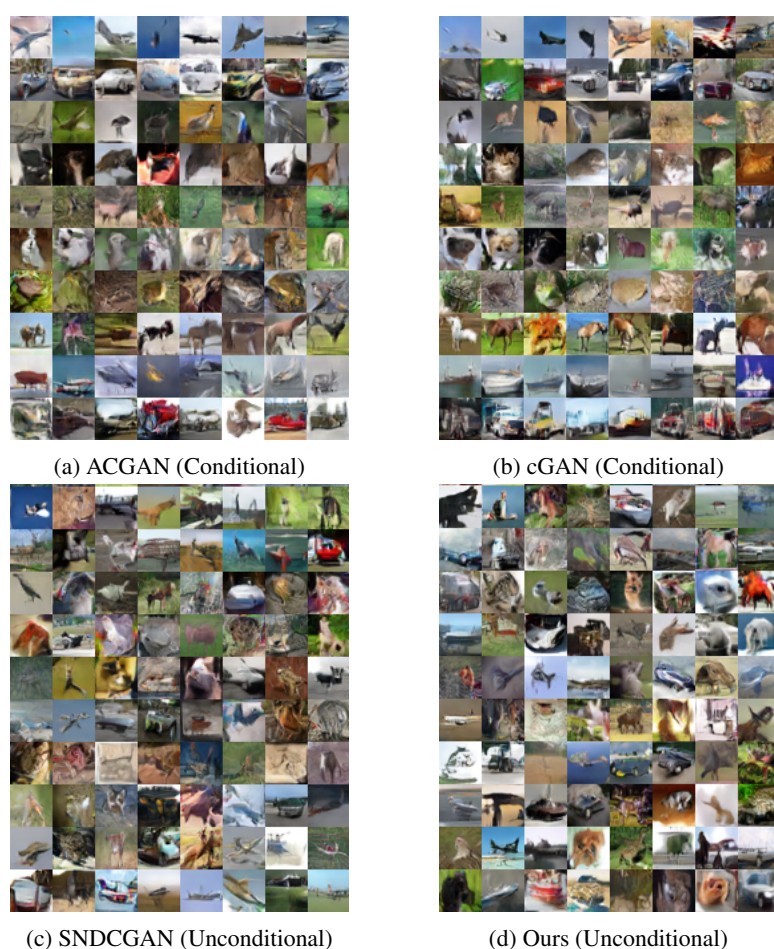

(a) ACGAN (Conditional)

(b) cGAN (Conditional)

(c) SNDCGAN (Unconditional)

(d) Ours (Unconditional)

Figure 9: Images generated by different GANs for CIFAR-10 with imbalance ratio ($\rho = 10$).

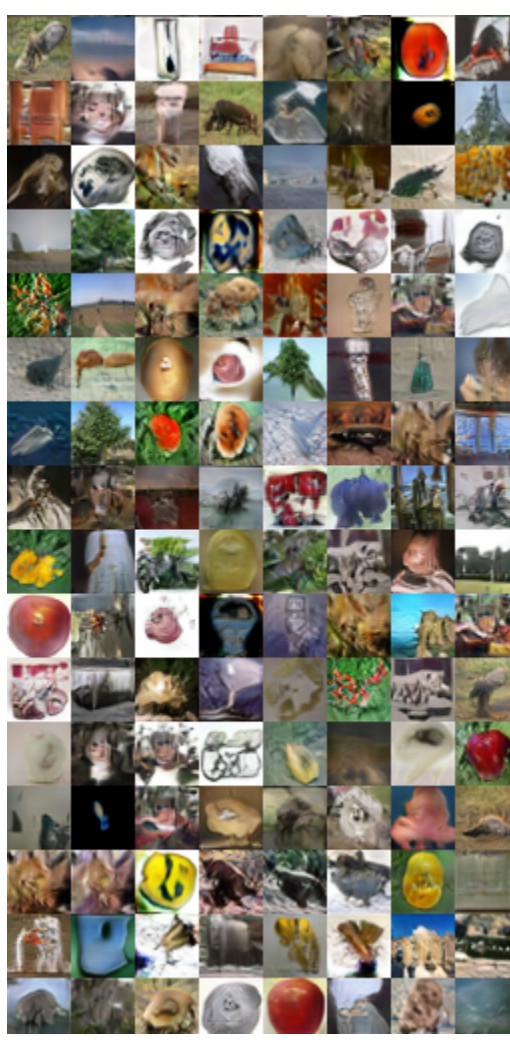

Figure 10: Images generated for CIFAR-100 dataset with our method (GAN + Regularizer).

