# OpenReview forum: "Class Balancing GAN with a Classifier in the Loop"
_ICLR.cc/2021/Conference — Reject_

### Official Review · AnonReviewer1 · 2020-10-18
**Interesting Simple Idea but not Convinced about Motivation and Results**

**Rating:** 4
**Confidence:** 4

**Review:**

**Overview**: The paper presents a simple regularizer term that aims to force a GAN to generate samples following a uniform distribution over different classes. The regularizer depends on a classifier that works well on an imbalanced or long-tailed dataset. The paper presents experiments on CIFAR-10 and LSUN that were synthetically long-tailed or imbalanced. The results show that the proposed term generates samples that follow a more uniform distribution over classes.

*Pros*:
- Interesting idea as it can help a generative algorithm to remove an imbalance from a dataset.
- The proposed regularizer is simple but depends on a classifier (see below for more details).

*Cons*:
- The regularization term depends on a classifier that works well already on the imbalanced dataset. Getting a classifier to work on long-tailed datasets is not an easy task and people are still investigating the development of techniques to learn from imbalanced datasets (see for example I).  From a practical point of view, this is a hard requirement that can reduce the chances of adoption.

- Proposed loss may have conflicting terms. The final loss composed of the relativistic loss and the regularizer may be conflicting. According to the text (below Eq. 3), this loss follows the training distribution which in the context of the paper is long-tailed. However, the proposed regularizer penalizes the GAN to generate samples following a long-tailed distribution. Aren't these two terms then conflicting? If so, can this conflict impact the convergence of the network?

- Insufficient experiments. While the experiments show good results on two small and synthetically long-tailed datasets, it is unclear if this method can work on naturally long-tailed datasets (e.g., iNaturalist). Unfortunately, the CIFAR-10 and LSUN datasets have a small set of classes in them. How does this method work on naturally long-tailed (e.g., iNaturalist) and/or large-scale datasets with a larger set of classes (e.g., ImageNet-LT)? Also, how do the generated images look like? Does this method still preserve a good perceptual image?

- Lack of clear impact on applications. After reading the introduction, I did not have a clear application where this component can be crucial to either enable a new application or solve a bottleneck. The discussion section briefly mentions a few applications. However, I think the paper would've been stronger if it showed experiments using the proposed approach and showing its impact on a clear application.

References:
I. Liu et al. Large-Scale Long-Tailed Recognition in an Open World. CVPR 2019.

Minor comments:
1. The contribution list of the Introduction section uses terms that have not been defined, i.e., FID and UAP.
2. If using latex, please use \min, \max, \log to properly display the operators.

----------------------------------------------------
Post Rebuttal Update

While I think the idea is interesting, I still think the proposed loss is not consistent as I still think the two terms in the loss collide with each other, its practical value is limited mainly because making a GAN to work on various datasets is a challenging task, and that the experiments now raised more questions than answers. For these reasons I still lean towards rejection as I believe the paper can benefit from a revision.

---

> ### Author Response · Authors · 2020-11-22
> **Response to Reviewer 1 (Part 1/2)**
>
> We are glad that the reviewer finds our method interesting. We provide clarifications to the questions asked below:
> *   Concern regarding the requirement of classifiers in the framework for hard imbalance datasets.
>     *   We do understand the reviewer’s concern about getting classifiers to work on long tailed distributions. In our experiments, we train the classifier also in the long-tailed setting using Deferred Reweighting[1]. Secondly, the progress on research in training classifiers on long-tailed data distributions is significantly better when compared to the progress on the front of training GANs on long-tailed distributions [1,2,3] . In our work we leverage the progress on classifier training to improve GAN training on long-tailed distributions.
>     *   Our method can also leverage available pre-trained classifiers on other similar datasets such as ImageNet, as is common practice. We show the results of our approach in a semi-supervised scenario, where we fine-tune an ImageNet pre-trained model using 0.1% labelled data in Section 4.2. Such approaches cannot be used in conditional GANs.
>     *   Conditional GANs (both cGAN and ACGAN) have a classifier integrated in the discriminator, which is crucial for training the generator. So, these architectures also suffer from the same class-imbalance issues as classifier training.
> *   Conflicting loss terms in the objective:
>     *   **Aren't these two terms then conflicting?** We believe there is no conflict in the two terms. The regularizer penalization is dependent on the current distribution of GAN generated samples ($N^t$) and does not focus on a particular set of minority classes throughout the training. The regularizer tends to increase the proportion of class k for which $N_k^t$ is lower (i.e. minority in current GAN  distribution). This adaptive phenomenon in regularizer ensures that GAN distribution becomes uniform. In the absence of the regularizer, the GAN would cater to the majority classes, and in such a case the GAN objective appears to be conflicting to the objective of the regularizer. However, in the presence of the regularizer, the GAN can generate images from minority classes. Since the primary objective of the Generator is to be able to fool the discriminator, this can be achieved on both majority and minority classes. Thus, the regularizer objective only aids the GAN towards generating minority class data, and does not go against the GAN objective.
>     *   **If so, can this conflict impact the convergence of the network?** We find similar convergence in FID values of GAN with and without the regularizer. We have provided the curve of FID vs number of iterations in Figure 5.
> *   Results on large datasets and perceptual quality:
>     *   Results on other datasets: In the revised paper (Table-3), we show results on long tailed CIFAR-100 dataset (Imbalance Ratio =10, SNResGAN architecture), where we are able to get better FID and also generate a balanced distribution similar to cGAN.
>     *   Rationale for choosing LSUN and CIFAR-10: It has been shown in existing works [4, 5] that the current GAN architectures work well on CIFAR-10 and LSUN. Since we aim to highlight a potential issue in the existing GAN implementations, we used long-tailed versions of the same dataset  to bring out the issues in the long-tailed case.
>     *   In our UAP experiments (Section 4.3), we generate 128 x 128 images using a DCGAN on a 1000 class dataset. We show that we are able to generate 968 distinct classes using the proposed approach. We are the first to show that a data free method is able to surpass the state-of-the-art data-driven method on the ImageNet dataset.
>     *   iNaturalist contains a very large number ( > 4000) of classes and images, and to the best of our knowledge, there are no existing GAN papers which show Image Generation baseline results on this dataset. Similarly no baselines were found for Imagenet-LT as well. We could not show results on this dataset due to computational limitations.  We show the generation of a large number of classes for UAP experiments (on ImageNet) and also show that our method works for long-tailed CIFAR-100 dataset.
>     *   **_Does this method still preserve a good perceptual image?_** Yes, we find that our regularizer is able to generate more perceptual images compared to the baseline (without regularizer) across all datasets. We use FID to measure the quality and diversity in images. We have provided generated images in Figure 8, 9 and 10  in Appendix for LSUN and CIFAR-10 and CIFAR-100 datasets.

---

> ### Author Response · Authors · 2020-11-22
> **Response to Reviewer 1 (Part 2/2)**
>
> *   Lack of Applications
>     *   In the revised paper (Section 4.2), we demonstrate training a class-balanced GAN using 0.1% labeled data. This shows that using the proposed regularizer, we can leverage pre-trained classifiers (trained on other related datasets), which was not possible in ACGAN and cGAN. This reduces the labeled data requirement for GAN training and also ensures that all classes are learnt uniformly even in the presence of class imbalance in training dataset.
>     *   Fairness Application: In addition to the applications discussed in the paper, our method can be used in fairness applications as well, where the classifier gives feedback about a certain attribute to equalize it in generation. For example Equalizing the proportion of males and females in the generated distribution.
>     *   We request the reviewer to kindly consider our Data-Free UAP experimental results as well. This is an application of the proposed regularizer, which helps the generation of diverse class images for data-free UAP generation. Our proposed data-free method currently surpasses the state-of-the-art method which uses data.
> *   We thank the reviewer for the additional comments, which have been fixed in the revised version. The terms FID and UAP have been defined in the Abstract and Introduction.
>
> [1]Cao, K., Wei, C., Gaidon, A., Arechiga, N., & Ma, T. (2019). Learning imbalanced datasets with label-distribution-aware margin loss. In _Advances in Neural Information Processing Systems_ (pp. 1567-1578).
>
> [2] Kang, B., Xie, S., Rohrbach, M., Yan, Z., Gordo, A., Feng, J., & Kalantidis, Y. (2019, September). Decoupling Representation and Classifier for Long-Tailed Recognition. In _International Conference on Learning Representations_.
>
> [3] Tang, K., Huang, J., & Zhang, H. (2020). Long-tailed classification by keeping the good and removing the bad momentum causal effect. _Advances in Neural Information Processing Systems_, _33_.
>
> [4] Heusel, M., Ramsauer, H., Unterthiner, T., Nessler, B., & Hochreiter, S. (2017). Gans trained by a two time-scale update rule converge to a local nash equilibrium. In _Advances in neural information processing systems_ (pp. 6626-6637).
>
> [5] Gulrajani, I., Ahmed, F., Arjovsky, M., Dumoulin, V., & Courville, A. C. (2017). Improved training of wasserstein gans. In _Advances in neural information processing systems_ (pp. 5767-5777).

---

### Official Review · AnonReviewer3 · 2020-10-26
**The method part is not clearly written.**

**Rating:** 5
**Confidence:** 3

**Review:**

This paper focuses on the problem that GANs' poor performance in the imbalanced dataset and presents the class balancing regularizer for training GANs, encouraging the GAN to pay more attention to underrepresented classes.
They induced the class distribution information using a pre-trained classifier, and the regularize utilizes the class distribution to penalize excessive generation of samples from the majority classes, thus enforcing the GAN to generate samples from minority classes.

Pros:
+ The motivation is clear.
+ It seems to cite the relevant literature (that I know of) and compare it to reasonably established attacks and defenses.
+ Simple/directly applicable approach that seems to work experimentally, but

Cons:
- The method part is not easy to follow. My understanding is the effective class frequency is the cumulative number of generated samples for each class (with a discount factor) and normalizing it yields the distribution of the generated samples. But I didn't get the relationship between this distribution and the following regularizer. Could you comment on that?
- As far as I understand, the regularizer encourages all the classes to be balanced within each batch by maximizing the entropy. I am a little bit concerned about this setting, what if we use a small batch size so that the class distribution can be imbalanced within one batch.
- Only LSUN subset and CIFAR10, which include 5 and 10 classes respectively. I am wondering about the performance on large-scale imbalanced datasets like iNaturalist.


----
Updated:
Thanks to the authors for the provided extra experiments and clarifications. Some of my concerns (e.g., how the batch size affected the performance) have been diminished, but I do agree with other reviewers that more baselines should be included.

---

> ### Author Response · Authors · 2020-11-21
> **Response to Reviewer 3**
>
> We thank the reviewer for his comments and suggestions.  We provide answers to the questions raised by the reviewer and describe the changes we have made:
> *   Relationship between class distribution and the regularizer
>
>     * $
> \underset{\hat{p} }{\max}   \sum_{k} \frac{\hat{p}_k\log(\hat{p}_k)}{N_k^t}
> $
>
>  The class distribution of the generated samples is used as inverse weight in the above  regularizer term. The $N^t$ is class distribution of GAN samples in cycle t and $\hat{p}$ is approximation of batch distribution. If $N_k^t$, the probability of class-k in the current generated distribution is low,  then $\hat{p}_k$ (i.e concentration of class-k) for the batches in the next cycle is encouraged to be larger as it would yield an improved objective value.  Whereas for another class, which has a large share in the current distribution (large $N_k^t$), having a large $\hat{p}_k$ is not advantageous as it has a large denominator of $N_k^t$. We have updated the notation to make it more clear and also added additional explanation to describe the connection in Section 3.2.
>
>
>
> *   Similarity of Objective to Maximize Entropy and concern of small batch size.
>     *   The regularizer term mentioned in the above section is a weighted version of the entropy which is maximized. The weight for class k is the inverse of the class probability $N_k^t$ which is estimated by steps in Section 3.1. If we consider an extreme case where batch_size &lt; num_classes the entropy can still be maximized by generating a fixed set of (batch_size) number of classes in the training process and network can ignore other classes. But in our case due to the weight of $N_k^t$ an increased value of  weighted entropy objective can be obtained by generating more samples from classes which have low $N_k^t$. This allows GAN to shift its focus from a fixed set of classes to other minority classes, this shifting process continues till $N^t$ attains uniform distribution. We provide results for DCGAN in table below on CIFAR-10 (Imbalanced ratio = 10) with small batch size using the same hyperparameters below,  the low KL Div of GAN class distribution to uniform show that the balancing effect is still preserved for lower batch sizes:
> | Batch Size 	| FID            	| KL DIV      	|
> |------------|----------------	|--------------	|
> | 16         | 50.05 +/- 0.15 	| 0.03 +/- 0.0 |
> | 32         | 38.58 +/- 0.04 	| 0.01 +/- 0.0 |
> | 256       | 30.48 +/- 0.07 	| 0.01 +/- 0.0 |
> Our results also show the same trend of bigger batch size leading to smaller FID values as seen in [3], when trained for a fixed number of iterations. We also request reviewer to refer to our UAP results in Sections 4.3  where we also train a GAN with our regularizer, with batch size of 512 to generate samples from 968 diverse classes.
>
>
> *   Performance on large-scale imbalanced datasets
>     *   Results on other datasets: In the revised paper (Table-3), we show results on long tailed CIFAR-100 dataset (Imbalance Ratio =10, SNResGAN architecture), where we are able to get better FID and also generate a balanced distribution similar to cGAN.
>     *   Rationale for choosing LSUN and CIFAR-10: It has been shown in existing works [1, 2] that the current GAN architectures work well on CIFAR-10 and LSUN. Since we aim to highlight a potential issue in the existing GAN implementations, we used long-tailed versions of the same dataset  to bring out the issues in the long-tailed case.
>     *   In our UAP experiments (Section 4.3), we generate 128 x 128 images using a DCGAN with a batch size of 512. We show that we are able to generate 968 distinct classes using the proposed approach. We are the first to show that a data free method is able to surpass the state-of-the-art data-driven method on the ImageNet dataset.
>     *   iNaturalist contains a very large number ( > 4000) of classes, and to the best of our knowledge, there are no existing GAN papers which show Image Generation baseline results on this dataset. We could not show results on this dataset due to computational limitations.  We show the generation of a large number of classes for UAP experiments (on ImageNet) and also show that our method works for long-tailed CIFAR-100 dataset.
>
> [1] Heusel, M., Ramsauer, H., Unterthiner, T., Nessler, B., & Hochreiter, S. (2017). Gans trained by a two time-scale update rule converge to a local nash equilibrium. In _Advances in neural information processing systems_ (pp. 6626-6637).
>
> [2] Gulrajani, I., Ahmed, F., Arjovsky, M., Dumoulin, V., & Courville, A. C. (2017). Improved training of wasserstein gans. In _Advances in neural information processing systems_ (pp. 5767-5777).
>
> [3] Brock, A., Donahue, J., & Simonyan, K. (2018, September). Large Scale GAN Training for High Fidelity Natural Image Synthesis. In _International Conference on Learning Representations_.

---

### Official Review · AnonReviewer2 · 2020-10-28
**Questionable significance of the presented results**

**Rating:** 5
**Confidence:** 4

**Review:**

Paper summary:

The paper proposes a regularizer to force an unconditional GAN generator to produce samples that follow a uniform class distribution. To provide feedback to the generator about the class distribution over the generated images, the proposed method utilizes a pretrained classifier on the same (imbalanced) training dataset. Motivated by the exponential forgetting of earlier tasks in neural networks [1], the regularization term encourages the generator to increase the proportion of samples of an infrequent class after a certain number of iterations and vice versa. Empirical studies are performed to show the effectiveness of the regularization: 1) the paper shows that the proposed method enables generating samples with a uniform class distribution with a GAN trained on a dataset with a long-tailed class distribution and (2) that the method benefits in generating universal adversarial perturbations (UAPs) in the data-free scenario.

Pros:

1.	The paper studies an important and challenging problem of training GANs on imbalanced dataset.
2.	The proposed regularization term is novel, the derivation of the regularization term is well explained.

Cons:

1. Image Generation

-	It is not clear whether it’s necessary to obtain a set of balanced samples in such a complicated way. Given a pretrained classifier as used in the proposed method, we could simply use the classifier to select samples after training a standard unconditional GAN. This simple baseline experiment is missing in the paper.

-	The experimental setup in Section 4.1 might be unfair for the unconditional GAN baselines. If the model is trained on an imbalanced dataset and tested on a balanced dataset, then the comparison between the proposed method and unconditional GANs is not fair. FID for the latter might be worse simply because the training and test distributions are different, whereas the proposed method is tailored for this special purpose.

-	The paper highlights that the method can generate images with uniform class distribution even in the unconditional case, where no labels are given to the generator. This would be useful if the classifier training is decoupled from the training data of the generator (this is not the case in the presented experiments, as the classifier uses the same training data as the generator with GT class labels). E.g. the classifier is trained on one dataset and then transferred for GAN training on another similar but unlabelled dataset, producing sharp images, or if only part of the training data was labelled, thus reducing the need for labels. Such experiments would help to support the claims in the paper. After all, the method is quite interesting, since in contrast to a conventional conditional GAN, the discriminator is not provided with the class identity of the generated image directly.

- The method and evaluation heavily relies on the quality of the pre-trained classifier. In the presented experimental setup, the classifier is trained on the same training set as the GAN model. So in case of the highly imbalanced training set, it's not clear how well the classifier can recognize the imbalanced classes. Thus the proposed model might suffer from the same problem as the unconditional GAN if the classifier has troubles recognizing imbalanced classes and is biased towards well represented classes.

- The problem posed in the paper is that unconditional GANs do not sample images from the present classes uniformly, but are biased by the class distribution in the dataset. This makes sense in the unconditional case, but Figure 2 shows that this also happens in the conditional case for ACGAN. Some further explanation for why that is would be helpful. Figure 2 is not discussed in the text, but it should be.

-	The produced samples are uniformly classified into different categories by the classifier. It would be good to provide a figure with generated images and the corresponding predicted labels, to see if the prediction and actual content match as well as in the case of a conditional GAN, like cGAN.

-	The proposed method is also trained with a big batch size of 256, which is very helpful for covering all classes. It would be more useful to see if the method also works well with small batch sizes of 16 or 32, which are common for high resolution GAN image synthesis.

-	Both the conditional GANs and the proposed regularizer use the same labelled data.

-	The paper says that in the highly imbalanced case cGAN sufferes from the training instability. In this case how is the batch formed for cGAN training? Do you balance the batch in terms of classes (as the batch size of 256 is quite large)? If not, it would be interesting to see how cGAN performs with the balanced batch.

2. Universal Adversarial Perturbations

-	The experimental results in Section 4.2 does not look convincing. The listed methods have different degrees of available information (either overlaps with the target dataset or a pretrained classifier on the target dataset). Moreover, for the proposed method, the requirement of a pretrained classifier on the target dataset is a strong limitation.

-	The paper claims that the method can help generate UAPs in the absence of the target dataset for which a classifier shall be fooled. The target dataset is ImageNet. The dataset on which the auxiliary classifier for the proposed regularization loss is trained is also ImageNet. Hence, it seems the UAPs could have been learned from ImageNet directly. To avoid confusion, sentences like the following need more elaboration: "We also find that our data free results are at par with the recently published method (Zhang et al., 2020) which uses ImageNet training data."

-	The following sentence needs more explanation: "Our approach achieves diversity through sampling from multiple checkpoints, as in each cycle the regularizer encourages the GAN to focus on different poorly represented classes". If checkpoints from different training steps are used, it seems that the target distribution is not really captured uniformly, contrary to the aim of the paper. Hence, it would be good to add some explanation here to prevent misunderstandings.

-	Table 2 is not discussed in the text.

Minor comments:

-  The height-width ratio of Figure 1b should be rectified.
-  The discussion is not well written. Especially the message of the 2nd bullet point needs more clarifications.


Review summary:

The paper shows that the classification and image generation can be decoupled for GANs. This makes the setup semi-supervised, not unconditional (as claimed in the paper). While this is interesting, it would be good for the paper to show an application where this is actually useful, because in all provided examples the ground truth labels of the training set are present and used for training the classifier, but are just not used directly to train the discriminator. For example, when it comes to UAPs it is not clear to what extent the proposed approach is data-free if ImageNet is used for training the regularizer as well as the classifier to be fooled. Thus, the main weaknesses of the paper in my opinion are the significance of the presented results, unfairness in the experimental setup, and the clarity of presentation.


Post-rebuttal feedback:

Thanks to the authors for the provided extra experiments and clarifications. I feel that my concerns have been partially addressed, thus raising my score to 5. I still think that the proposed method is limited by the classifier, it's ability to capture a long-tailed distribution, which is not so easy to get when trained on imbalanced dataset. This significantly limits the applications of the proposed approach in real-life scenarios. The paper has also experimented only on artificially created imbalanced datasets, which contain small number of classes with the model being trained with the batch size higher than the number of classes. It would be beneficial to see how the model would perform in more realistic setup when the number of classes is significantly bigger than the batch size (e.g. iNaturalist or even ImageNet), to support more the claims of the paper.

---

> ### Author Response · Authors · 2020-11-22
> **Response to Reviewer 2 (Part 1/2)**
>
> We thank the reviewer for his valuable comments and suggestions. We have tried to improve the paper on suggestions. We provide the clarifications to the concerns below:
>
> 1. Image Generation
> *   GAN Baseline of generating samples and using classifiers to provide labels.
>     *   From Figure 2 it can be seen that certain classes get mode collapsed in case of unconditional GAN and the distribution of samples learnt is arbitrary (different from long tailed distribution). If we consider a basic probability model of geometric distribution for getting 5k samples from minority class, assuming the distribution of generated samples as  in Figure 2. We would require an expected number of 1.2  million sampling steps to get 5k minority samples. This is not a principled/efficient way of generating samples.
> *   Unconditional GAN may have a lesser FID. (Unfairness)
>     *   Intuitively that was our expectation as well. But we find the converse being true which is an interesting observation. The FID of uncondtional GAN is better than conditional GAN in long-tail cases (in case of imbalance ratio = 100 as shown in Table 1) which is also described in Section 2.2. Similar observation is also made in the concurrent paper Data Efficient GANs [1] which shows conditional GANs suffer more when used in scarce data scenarios, inline with our experiments.
> *   Experimentation in semi supervised setup with the pre trained classifier trained using other sources.
>     *   We thank the reviewer for suggesting these experiments.  We have now added Section 4.2 in which we use a classifier which is obtained by fine tuning an ImageNet pretrained model with 0.1% of labelled data.  This classifier used in our GAN + Regularizer framework also provides balanced distribution compared to unconditional GAN. Unlike cGAN and ACGAN, which specifically requires labeled samples, our method doesn’t depend on the labels. All we need is a basic classifier with reasonable accuracy.
> *   Reliance of  the method and evaluation on a pre-trained classifier which can overfit on majority classes.
>     *   We use a technique called Deferred Reweighting (which ensures all classes have reasonable performance) [1] for training classifiers on long tailed distributions which are used in GAN training framework. Also in the answer above, we show that our method is compatible with classifiers learnt using transfer learning too. We provide per class  accuracies for CIFAR10 long tail distribution (imb factor = 100) below: \
> Validation Accuracies:
>  [0.934,0.978,0.776,0.715,0.787,0.685,0.776,0.647,0.587,0.620]\
> The training samples decrease for each class as we go from left to right in an exponential fashion.
>     *  The classifier used for evaluation of all GANs is trained on a balanced dataset and has a higher validation accuracy (see Appendix A.3). This classifier is only used for evaluation and is not used in regularizer formulation for GAN training.
>
> *   Explanation for ACGAN having a biased distribution:
>     *   ACGAN loss consists of two loss terms i.e. GAN Loss (Real/Fake) + Classification Loss. As the GAN distribution is imbalanced the discriminator tends to classify majority class images as real and ignore other classes. When a fake class label is given to a generator to decrease the loss it has two options  one is to decrease the GAN loss (Real/Fake) by generating majority class or reduce the classification loss by generating an image of correct class. In such cases sometimes the generator favours to decrease GAN (Real/Fake) loss and ignores the label to generate majority samples which leads to imbalance.
> *   Figures for samples with labels.
>     *   We share images of LSUN dataset (Imbalance Ratio = 10) with labels from the classifier. [http://s000.tinyupload.com/?file_id=21497176106115832192/](http://s000.tinyupload.com/index.php?file_id=21497176106115832192). The validation accuracy on balanced validation set for different classifiers used is present in Appendix A.3
> *   The proposed method is also trained with a big batch size of 256, which is very helpful for covering all classes. It would be more useful to see if the method also works well with small batch sizes of 16 or 32, which are common for high resolution GAN image synthesis.
>     *   We provide results for DCGAN on CIFAR-10 (Imbalanced ratio = 10) with small batch size using the same hyperparameters below,  the low KL Div of GAN class distribution to uniform show that the balancing effect is still preserved for lower batch sizes:\
>  | Batch Size 	| FID            	| KL DIV      	|
> |------------	|----------------	|--------------	|
> | 16         	| 50.05 +/- 0.15 	| 0.03 +/- 0.0 	|
> | 32         	| 38.58 +/- 0.04 	| 0.01 +/- 0.0 	|
> | 256        	| 30.48 +/- 0.07 	| 0.01 +/- 0.0 	|\
> Our results also show the same trend of bigger batch size leading to smaller FID values as seen in [3], when trained for a fixed number of iterations.

---

> ### Author Response · Authors · 2020-11-22
> **Response to Reviewer 2 (Part 2/2)**
>
> 1. Image Generation (Continued from previous part sequentially)
> *   Both the conditional GANs and the proposed regularizer use the same labelled data.
>     *   Yes in our current experiments that is the case. But the performance achieved by our method is superior to the performance by cGAN(which is the only one producing balanced distribution). We have achieved better FID in 3 out of 4 cases. Our method is able to get better downstream classifier accuracy in all cases  as shown in Table 1. Also we have now provided results with a classifier trained with 0.1 % labeled data in Section 4.2.
> *   Balanced Batch for cGAN training.
>     *   We tried the method of balanced resampling and found that the instability issue still persists and we get a FID of 56.89 +/- 0.04 which is worse than using cGAN (FID 48.13 +/- 0.01) without resampling (which we initially used).
> 2. Universal Adversarial Perturbations (UAP)
> *   Unfairness in comparison of UAP Results
>     *   In the data free approaches it is assumed that the classifier is available. All the compared methods (in Table 4) make use of a classifier + some prior data which is not ImageNet.
>
>         GDUAP + P - We report GDUAP results with prior texture data of texture obtained from the paper [2].
>
>         PDUA + P - Method uses classifier + some prior texture data
>
>         AAA - Method uses classifier + Prior data obtained from activation Maximization
>
>         MI ADV - Classifier + Prior Data COCO (This is highlighted as the prior data COCO overlaps with Imagenet which can make it less challenging to craft the perturbations)
>
>        Our method- Uses classifier (In Regularizer and UAP algorithm) + Prior Comics Data
>
>     *   Data Free UAP practicality: Lots of deep learning models are provided to user devices without releasing the training data due to privacy and other concerns. Also the datasets are huge in size which attracts significant overhead to handle them which is not required in case of data free methods making it efficient for the attacker. So Data Free methods are helpful for adversary creation on models in  the above cases.
> *   UAP can be learnt on ImageNet data and clarification of comparison of results.
>     *   In the data free method we assume access to the trained classifier which has to be fooled. The training data on which it is learned is not available. In this case, the attacker may use arbitrary or proxy data samples for crafting the perturbations. It is usually considered that attacks which don’t use ImageNet data are weaker then the attacks created using ImageNet data. To show that our data-free method is comparable to cases when an attack is created using ImageNet data we had added the last row on Table 4. We have now clarified the statement by adding the exact difference in results. We thank you for the suggestion.
> *   Sampling using multiple checkpoints in case of DCGAN
>     *   The aim of the regularizer is to shift the focus of GAN towards generation from different minority classes. Due to the limited capacity of DCGAN it is bound to forget modes and shift to new different minority classes (which is the aim of the regularizer). Hence we sample from multiple cycles to cover all classes, For generation of all ImageNet classes simultaneously, large GANs are required. This limited capacity issue is also observed in ACGAN paper where the authors use 100 DCGANs to generate 1000 ImageNet classes. We have added the explanation in the revised paper.
> *   Table 2 is not discussed in the text.
>     *   We have now discussed both Table 2 and Table 3 in the text.
> Minor comments:
> *   The height-width ratio of Figure 1b should be rectified.
>     *   We have fixed the height to width ratio.
> *   The discussion is not well written. Especially the message of the 2nd bullet point needs more clarifications.
>     *   We have clarified the 2nd point by performing experimentation in semi supervised setup.
>
> *   GAN + regularizer is a  semi-supervised setup, not unconditional (as claimed in the paper).
>     *   By an unconditional model we refer to the fact that our GAN is still of the form G(z) and does not require a label for generation of sample. Whereas conditional GANs model G(z|y) which requires a label for generation of image. Yes, our method can be used in both supervised and semi supervised settings.
>
> [1] Zhao, S., Liu, Z., Lin, J., Zhu, J. Y., & Han, S. (2020). Differentiable augmentation for data-efficient gan training. _Advances in Neural Information Processing Systems_, _33_. \
> [2] Liu, H., Ji, R., Li, J., Zhang, B., Gao, Y., Wu, Y., & Huang, F. (2019). Universal adversarial perturbation via prior driven uncertainty approximation. In _Proceedings of the IEEE International Conference on Computer Vision_ (pp. 2941-2949). \
> [3] Brock, A., Donahue, J., & Simonyan, K. (2018, September). Large Scale GAN Training for High Fidelity Natural Image Synthesis. In _International Conference on Learning Representations_.

---

### Official Review · AnonReviewer4 · 2020-10-28
**The idea is intuitive and hope to have more experiments**

**Rating:** 5
**Confidence:** 4

**Review:**


This paper tries to solve the data imbalance problem in conditional GAN by adding a classification loss as the constraint. This constraint can be seen as weighted softmax and the weight is the smoothed number of classes showed before.

This paper is well written and structured. The idea is intuitive. The authors applied the methods in long-tail classification to GAN. In the experiment part, the authors first did the image generation from long-tailed distribution on CIFAR10 and LSUN, the apply this technique to data-free universal adversarial perturbation.  The results are better than the ones without the proposed constraints.

My main concern is that the dataset used in the first part is too simple to reflect the strength of the algorithm. Since the method is very intuitive, more experiments on datasets with larger pixel number and more categories are more convincing.

Another concern is the baseline. Since current state-of-art conditional GAN can reach lower FID (compare with FID score in the last column of Table 1), it might be better to compare with those methods. And there exist other intuitive baselines like data-augmentation and resampling.  is it fair to compare your methods with these?

---

> ### Author Response · Authors · 2020-11-21
> **Response to Reviewer 4**
>
> We thank the reviewer for the valuable comments.
>
>
>
> *   We would like to first clarify the significance of the proposed loss function, and highlight its difference with respect to a simple weighted classification loss. The loss function used in the paper is shown below:
>
>     $\underset{\hat{p} }{\max}  \; \sum_{k} \frac{\hat{p}_k\log(\hat{p}_k)}{N_k^t} $
>     *  Although the regularizer term in the above equation resembles the weighted loss in structure, the term is different from the weighted cross entropy (classification) loss. In the regularizer term the $\hat{p}_k$ is an approximation to the fraction of class-k samples in the batch. The distribution of the batch is encouraged to generate samples from classes which have lower $N_k^t$ (i.e. low concentration in generated output at a particular time t), and hence achieve a balanced distribution.
>  The weighted cross entropy (classification) loss requires ground truth labels for the training images. Contrary to this, in our approach we only require outputs from a pre trained classifier. We demonstrate the effect of regularizer though the experiments in Figure 1(b) and theoretical results in Proposition 1.
>
>
>
> *   Dataset too simple, need results on with larger pixel numbers and more categories
>     *   Results on other datasets: In the revised paper (Table-3), we show results on long tailed CIFAR-100 dataset (Imbalance Ratio =10, SNResGAN architecture), where we are able to get better FID and also generate a balanced distribution similar to cGAN.
>     *   Rationale for choosing LSUN and CIFAR-10: It has been shown in existing works [3, 4] that the current GAN architectures work well on CIFAR-10 and LSUN. Since we aim to highlight a potential issue in the existing GAN implementations, we used long-tailed versions of the same dataset  to bring out the issues in the long-tailed case.
>     *   In our UAP experiments (Section 4.3), we generate 128 x 128 images using a DCGAN with batch size 512. We show that we are able to generate 968 distinct classes using the proposed approach. We are the first to show that a data free method is able to surpass the state-of-the-art data-driven method on the ImageNet dataset.
> *   Issue with baseline, current SOTA reaches lower FID
>
>     * We would like to clarify that we use the DCGAN architecture for CIFAR-10 experiments. We share the mean FID score in the following table for comparison to recently published results [1, 2] with the same architecture but in a different hyperparameter setup:
> | Method             	| ACGAN    	| cGAN     	| SNDCGAN  	|
> |--------------------	|----------	|----------	|----------	|
> | Published Results  	| 21.44[1] 	| 19.52[1] 	| 27.50[2] 	|
> | Our Results        	| 24.21    	| 18.79    	| 27.05    	|
> State-of-the-art GANs use ResNet based large GANs which we use for LSUN experiments (in Table 1) to show the compatibility of the regularizer with both architectures.
>
>
>
> *   Comparison with data augmentation and resampling
>
> Resampling:
>
>
>
> 1. Resampling requires labels for the training samples, whereas our method requires only a classifier. We show that our method is also effective in semi semi-supervised setting with 0.1 % labels in Section 4.2 of revised paper.
> 2. We tried Resampling for cGAN as it was unstable (for CIFAR-10 with Imbalance Ratio = 100). It gives worse results (FID 56.89 +/- 0.04)  compared to the no resampling case (FID 48.13 +/- 0.01).
>
> Data Augmentation: Data augmentation forces the discriminator to learn better semantic features, which can be used with our method as well, to improve the results. We thank the reviewer for this suggestion, and we will investigate further on this.
>
> [1] ContraGAN: Contrastive Learning for Conditional Image Generation M Kang, J Park - _Advances in Neural Information Processing Systems_, 2020
>
> [2] Kurach K., Lučić M., Zhai X., Michalski, M., and Gelly S., A large-scale study on regularization and normalization in GANs. In _International Conference on Machine Learning_, 2019.
>
> [3] Heusel M., Ramsauer H., Unterthiner T., Nessler B., and Hochreiter S., Gans trained by a two time-scale update rule converge to a local nash equilibrium. In _Advances in Neural Information Processing Systems_, 2017
>
> [4] Gulrajani I., Ahmed F., Arjovsky M., Dumoulin V.,  and Courville A. C., Improved training of wasserstein gans. In _Advances in Neural Information Processing Systems, _2017.

---

### Comment · Area_Chair1 · 2020-11-22
**Discussion needed**

Dear Reviewers,

The authors have provided a detailed response and uploaded their revised manuscript. Would you please take a careful look at their response and revision? Please respond to the authors and update your review accordingly.

Thanks,
AC

---

### Decision · Program_Chairs · 2021-01-07
**Final Decision**

**Decision:**

Reject

**Comment:**

The authors have provided very detailed responses and added additional experimental results, which have helped address some of the referees' concerns. However, since the modification made to a vanilla GAN algorithm is relatively small, the reviewers are hoping to see the experiments on more appropriate real-world datasets (not artificially created imbalanced datasets with relatively few classes), more/stronger baselines, and rigorous theoretical/empirical analysis of the method's sensitivity to the quality of the pre-trained classifier. The paper is not ready for publication without these improvements.